# Evaluation of IAQ Management Using an IoT-Based Indoor Garden

**DOI:** 10.3390/ijerph17061867

**Published:** 2020-03-13

**Authors:** Ho-Hyun Kim, Min-Jung Kwak, Kwang-Jin Kim, Yoon-Kyung Gwak, Jeong-Hun Lee, Ho-Hyeong Yang

**Affiliations:** 1Department of Information, Communication and Technology Convergence, ICT Environment Convergence, Pyeongtaek University, 3825 Seodong-daero, Pyeongtaek-si 17869, Gyeonggi-do, Korea; 2Department of Data Information and Statistics in Pyeongtaek University, 3825, Seodong-daero, Pyeongtaek-si 17869, Gyeonggi-do, Korea; mjkwak@ptu.ac.kr; 3Urban Agriculture Research Division, National Institute of Horticulture and Herbal Science, 100, Nongsaengmyeong-ro, Iseo-myeon, Wanju-gun 55365, Jeollabuk-do, Korea; kwangjin@korea.kr; 4Life & Industry Environmental R&D Center in Pyeongtaek University, 3825, Seodong-daero, Pyeongtaek-si 17869, Gyeonggi-do, Korea; yoonkyung.gwak@gmail.com (Y.-K.G.); ljh5369@naver.com (J.-H.L.);

**Keywords:** internet of things (IoT), indoor gardens, indoor air quality, index

## Abstract

This study was designed to verify the effectiveness of smart gardens by improving indoor air quality (IAQ) through the installation of an indoor garden with sensor-based Internet-of-Things (IoT) technology that identifies pollutants such as particulate matter. In addition, the study aims to introduce indoor gardens for customized indoor air cleaning using the data and IoT technology. New apartments completed in 2016 were selected and divided into four households with indoor gardens installed and four households without indoor gardens. Real-time data and data on PM_2.5_, CO_2_, temperature, and humidity were collected through an IoT-based IAQ monitoring system. In addition, in order to examine the effects on the health of occupants, the results were analyzed based on epidemiological data, prevalence data, current maintenance, and recommendation criteria, and were presented and evaluated as indices. The indices were classified into a comfort index, which reflects the temperature and humidity, an IAQ index, which reflects PM_2.5_ and CO_2_, and an IAQ composite index. The IAQ index was divided into five grades from “good” to “hazardous”. Using a scale of 1 to 100 points, it was determined as follows: “good (0–20)”, “moderate (21–40)”, “unhealthy for sensitive group (41–60)”, “bad (61–80)”, “hazardous (81–100)”. It showed an increase in the “good” section after installing the indoor garden, and the “bad” section decreased. Additionally, the comfort index was classified into five grades from “very comfortable” to “very uncomfortable”. In the comfort index, the “uncomfortable” section decreased, and the “comfortable” section increased after the indoor garden was installed.

## 1. Introduction

Industrialization and urbanization cause environmental deterioration and a reduction in green space, while the encapsulation of buildings to improve energy efficiency threatens the health of the occupants, even in indoor environments [1]. As the influence of indoor air quality on the human body has been gradually revealed, O_3_, NO_2_, SO_2_, and PM_10_ have been identified as indoor pollutants affecting human life and health and have been regulated by environmental standards [2]. Among them, particulate matter was found to contribute to the high rate of early mortality since it affects the respiratory system adversely by penetrating the respiratory organs [3,4]. Particulate matter was defined as a Group 1 carcinogen in 2013 [5]; among them, PM_2.5_ was designated as a Group 1 carcinogen by the World Health Organization (WHO), since it not only causes respiratory disease but also enters the brain directly through the olfactory cells, potentially causing dementia [6]. In addition, the particulate matter generated has been found to affect human disease and mortality [7,8,9,10]. Another pollutant, CO_2_, rather than causing illness on its own, may affect the human body by causing metabolic disorders such as sick building syndrome when a person is exposed to high concentrations [11]. In order to reduce indoor air pollution, a method using a plant-based system is emerging as an efficient technique [12]. In fact, foliage plants (*Ficus Benjamina, Chamaedorea elegan, Ficus elastic*, etc.) that are widely used indoors have been shown to be effective at removing CO_2_ [13]. In addition, 10 foliage plants (*Hedera helix, Nephrolepis exaltata, Epipremnum aureum, Davallia mariesii*, etc.) and three species of hanging Tillandsia (*Tilandsia usneoides*, etc.) have been found to be effective at removing PM_10_ and PM_2.5_ [14].

The Internet-of-Things (IoT) is considered a new technology through which devices are developed for improving everyday activities [15] and monitoring environmental parameters in various areas. Wireless sensors have been developed for indoor air-quality monitoring and to provide real-time information. Furthermore, wireless sensors collect big data and offer important opportunities for improving indoor air quality with applications such as indoor air-quality control [16].

Recently, as particulate matter warnings and alarms are frequently issued in the Republic of Korea, interest in exposure to and control of particulate matter has increased. In addition, in line with the Fourth Industrial Revolution era, real-time air-quality sensing evaluation technology based on sensors for particulate matter with fast processing capacities at relatively low prices, in conjunction with sensor-based Internet-of-Things (IoT) and secured data representativeness through big data analysis, has become a feasible option. These results enable the creation of special applications of IoT through big data-based data analysis in which large amounts of data are collected and analyzed. Air-quality monitoring, or big data, formed in this way will alert the occupants in indoor spaces when air-quality management is required or will help with the determination of the best ventilation system in the indoor spaces [17]. In addition, in recent years, social demands for the application of a smart indoor garden incorporating IoT technology have also been increasing. That is, interest in the introduction of green spaces and the purification of indoor air quality is increasing in sensitive group facilities, such as underground spaces like subway stations, daycare centers, kindergartens, and schools. In addition, there may be viable solutions to gardening issues based on the application of IoT technology in various situations through previous research or experiments [18].

Previous methods of measuring the effectiveness of indoor plants had difficulties in assessment. The first reason was particulate matter measuring devices could cause inconvenience to people due to the noise and the space occupied. In addition, there were limitations related to the comparison of seasonal data because the gravimetric method had difficulties extracting data. In this study, it was verified that small IoT devices could be installed in multiple units in households which produce real-time data.

Therefore, the purpose of this study was to analyze the effect of indoor environment control of particulate matter and carbon dioxide using plant applications by creating a database for indoor adaptability and environmental requirements of plants based on an efficient plant-based air purification system and by utilizing the data generated using smart sensors for particulate matter, one of the core elements of IoT. This study also aimed at evaluating the adequacy of smart indices for the air purification indoor garden integrated solution using the analyzed data.

## 2. Materials and Methods

This study monitored eight new apartments in Pyeongtaek City from June 2017 to September 2018. An IoT-based indoor air-quality (IAQ) monitoring system, including remote sensing (PM_2.5_, CO_2_, temperature, and humidity), was established after dividing the apartments into four households with indoor gardens installed and four households without indoor gardens installed. The eight households had the same amount of floor space (116 m^2^) in the same apartment complex, the construction of which was completed in December 2016. The characteristics of all households are shown in Table 1.

### 2.1. Sensor Installation and Information

Sensors were installed in the living room and kitchen of each household to measure PM_2.5_, CO_2,_ temperature, and humidity every five minutes. In addition, an exterior device was installed on the outer walls of the households with indoor gardens installed to identify the external environment of the new apartment building (Figure 1). The IoT device sensor used in the installation was utilized for this purpose (Table 2).

Honeywell’s HPM Series Particle Sensor is a laser-based sensor using a light-scattering method with a particulate matter measurement range of 0 to 1000 µg/m^3^. The laser light illuminates the particulate matter through the detection chamber in which plants absorb particulate matter. The particulate matter passes through the laser beam, and the light is recorded in the light detector or in the picture. This light is analyzed, and the concentration is calculated in real time to convert it into an electronic signal.

The device, which is capable of stable operation and continuous use for 20,000 hours, has an accuracy of error range of ±15 µg/m^3^ and can make measurements without further calibration.

### 2.2. Indoor Garden Installation

In the case of the households with indoor gardens installed, the indoor garden was installed after collecting basic data using the sensor inside the household without the plants for about a month before installing the indoor garden, and plants were placed around 3% of the area of the kitchen and living room. Five large pots, two Bio Walls, and four Wiz-Pots were installed, and the areas before and after the installation of the indoor garden were compared. The dimension of the indoor garden was 1.2 (L) × 0.3 (W) × 1.8 (H) in the living room and 1 (L) × 0.3 (W) × 1.2 (H) in the kitchen (Figure 2).

Among the installed plants, *Ficus* (*Ficus elastic* and *Ficus benghalensis*) and *Rhapis excelsa* are effective at removing formaldehyde, xylene, and toluene, while *Fragrant Aralia* and *Schefflera arboricola Hayata* are known to be effective at purifying air [19]. Afterwards, changes in indoor air quality before and after garden installation were analyzed. The background concentration of each household was measured within three months.

### 2.3. Calculation of an Indoor Environment Comprehensive Indoor Air-Quality (IAQ) Index

The indoor environmental standards are based on the current maintenance standards and recommended standards and are divided into five grades based on effects on humans [20,21,22], as shown in Table 3 below.

Index sections of “unhealthy for sensitive groups” are set based on the health effects by concentrations such as standard epidemiological data including indoor environmental standards, and the sections are calculated by considering the background concentration, exposure frequency, and compliance of each indoor space. In terms of the trends in domestic and foreign environmental index scores, the same concept was utilized in this study, considering that the higher the score, the more polluted the air quality. Therefore, it was determined that, using a scale of 1 to 100 points, the higher the score, the worse the indoor air quality and the higher the risk to the human body.

The calculation of the PM_2.5_ index, in Section 2.3.1, the calculation of the CO_2_ index in Section 2.3.2 and the final calculation of the indoor environment comprehensive in the IAQ index section, a detailed classification of A–E is shown in Section 2.3.3.

#### 2.3.1. Calculation of the PM_2.5_ Index

In the case of PM_2.5_, the indices were considered with respect to the prevalence effects on humans. For the index section of “unhealthy for sensitive groups,” the main target group in this study, an integrated index section was set within the concentration range of 120 µg/m^3^, the level of a 10% increase in the prevalence of acute chronic obstructive pulmonary disease (COPD) in the public, and based on 80 µg/m^3^ United States Environmental Protection Agency (US EPA) Air Quality Index, AQI), the level of a 10% increase in the prevalence of COPD in children. “bad” and “hazardous” labels were applied based on particulate matter warnings and alarm standards for the standard domestic atmospheric. The upper limit on “hazardous” has no data on the effects on humans, and the upper limit of the US EPA AQI is estimated to be 600 µg/m^3^.

#### 2.3.2. Calculation of the CO_2_ Index

For CO_2,_ “good (0–450 ppm)” to “moderate (451–700 ppm)” values were determined using the relative risk data obtained from the epidemiological data, and “moderate (451–700 ppm)” was the level to cause a wheeze. The “unhealthy for sensitive groups (701–1000 ppm)” was based on the effects baseline among relative risks, American Society of Heating, Refrigeration, and Air-Conditioning Engineers (ASHREA) long-term recommended levels (700 ppm) and indoor environmental standards cut-off level (1000 ppm). Furthermore, the “bad (1001–3000 ppm)” to “hazardous (>3000 ppm)” was based on high-concentration human impact data among the ASHREA data.

#### 2.3.3. Final Calculation of the Indoor Environment Comprehensive IAQ Index Section

In this study, Table 4 shows the comprehensive presentation (proposal) of the index sections of good, moderate, unhealthy for sensitive groups, bad, and hazardous levels via assessment items (PM_2.5_, CO_2_) of the indoor environment comprehensive IAQ index; the colors for the classifications are given in Table 5.

After dividing the five classes equally, using a scale of 0 to 100 points, it was determined that the higher the score, the lower the IAQ, which considers comfort, such as temperature and humidity, and the safety of indoor harmful substances. The index is shown in blue when the indoor environment is good for the sensitive groups, green when the level is good for the general public, and yellow when the level could have a slight effect on the sensitive groups. The discomfort and health effects for sensitive groups are indicated in orange, and the level of discomfort and health effects for all occupants are marked in red.

### 2.4. Comfort Index Calculation Criteria

In order to calculate the comfort index required for the calculation of the IAQ composite index, the basic data on the comfort temperature and comfort humidity index using the data results of the indoor air measurement sensor were used. The total score of the indices was 100 points. The index was divided into static variables (maximum, minimum, median) from survey data and input variables from measurements. The range of the comfort index was calculated for temperatures from 20 to 26 °C and humidity from 30 to 70%.
(1)$$\frac{1}{I}\overset{I}{\underset{i=1}{\sum }}100\left(\frac{\left|C_{i}^{\mathrm{opt}}{-C}_{i}^{\mathrm{obs}}\right|}{C_{i}^{\max}{-C}_{i}^{\min}}\right)$$
$$C_{i}^{\mathrm{opt}}=\mathrm{Median}\;\mathrm{of}\;\mathrm{comfort}\;\mathrm{index}$$
$$C_{i}^{\mathrm{obs}}=\mathrm{Measuring}\;\mathrm{value}$$
$$C_{i}^{\max}=\mathrm{Maximum}\;\mathrm{of}\;\mathrm{comfort}\;\mathrm{index}$$
$$C_{i}^{\min}=\mathrm{Minimum}\;\mathrm{of}\;\mathrm{comfort}\;\mathrm{index}$$
$$C_{i}{=C}_{T}{,C}_{R}{,C}_{L}{,C}_{N}$$

The comfort index was divided into five grades considering the development of the comprehensive index of comfort and safety, which is shown in Table 6.

The score range of the comfort index (temperature and humidity) in this study was set to align with the tendency of the domestic and international environment-related index scores so that the higher the score, the more polluted the air quality. The points were classified into colors using a scale of 0 to 100 points, showing that the higher the score, the less comfortable the environment. It is set to blue when the sensitive group feels comfortable in the indoor environment, green when the level of comfort is felt by the general public, and yellow when the sensitive groups feel slight discomfort. The index is shown in orange when the indoor environment is uncomfortable to the sensitive groups and is shown in red when all the occupants are uncomfortable.

### 2.5. Development of an IAQ Composite Index (Proposal)

The development of the IAQ composite index (IAQ index + comfort index) (proposal) was classified into five grades used in each index for the comfort index (proposal) and IAQ index (proposal) proposed in this study, as shown in Table 7. Using the scale of 1 to 100 points, it was determined that the higher scores represent a lower quality of the IAQ and the comfort indices, as indicated by the temperature, humidity, and safety.

The IAQ composite index for indoor gardening was calculated by the following two factors:(2)$$Index=\omega_{1}\times IAQ\;index+\omega_{2}\times comfort\;index$$

The IAQ index, which is composed of two components, PM_2.5_ and CO_2_, and the comfort index, which is composed of temperature and humidity, are calculated on a scale of 100 points. The components ω_1_ and ω_2_ can be expressed according to the following equations when the weights of the IAQ index and the comfort index are calculated as 1:1 in the determination of the composite index:(3)$$\begin{array}{c} Compostie\;index=0.5\times IAQ\;index+0.5\times comfort\;index\\ =\frac{1}{2}\left(PM_{2.5}+CO_{2}\right)Index+\frac{1}{2}comfort\;index \end{array}$$

### 2.6. Data Analysis and Index Calculation

On the basis of the real-time and long-term collected data, the changes in PM_2.5_, CO_2_, temperature, and humidity according to the installation of indoor gardens were compared and analyzed. The results of the analysis based on epidemiological data, prevalence data, current maintenance, and recommendation criteria are presented as indices to increase the visibility of the analyzed results and to provide information on how the current indoor air quality can affect the health of the occupants. The index was classified into a comfort index, which reflects temperature and humidity, an IAQ index that reflects PM_2.5_ and CO_2_, and an IAQ composite index. The IAQ composite index was calculated by combining the comfort index and the IAQ index.

## 3. Results

### 3.1. Measurement and Data Analysis

On the basis of the data measured during the investigation, the data statistics for each household were tabulated (Table 8). The range of PM_2.5_ in the installed households ranged from 16.8 to 31.8㎍/m^3^ before installation and 13.8 to 24.6 µg/m^3^ after installation. In addition, the households without gardens installed ranged from 8.2 to 35.5 µg/m^3^. There was no clear difference in the concentrations between the garden-installed and non-installation households since the average concentration of PM_2.5_ had low distributions due to the characteristics of the household. On the other hand, the before and after reviews of the indoor garden installation in the garden-installed households showed a change in the concentration of particulate matter. After the installation, the concentration of the particulate matter was found to have decreased.

The range of CO_2_ was 653.9 to 1185.4 ppm before installation and 570.0 to 1141.4 ppm after installation in the garden-installed households. In the non-installation households, it was 741.8 to 1012.8 ppm. Comparing the areas before and after the installation of the indoor garden, the concentration of CO_2_ was reduced after the indoor garden was installed.

There is no standard for PM_2.5_ and CO_2_ in the indoor air-quality recommendation standards of new apartment buildings. Accordingly, when comparing these values to the indoor air-quality maintenance standard (PM_2.5_ 35 µg/m^3^ (sensitive class facility); CO_2_ 1000 ppm) of the Indoor Air Quality Law regarding multi-use facilities, all PM_2.5_ values were lower than the standard, and the CO_2_ concentrations were greater than 1000 ppm in the B households, while the concentration decreased after installing the gardens (Korean Legislation Research Institute, 2017).

The humidity ranged from 32.7% to 40.5% before installation and 37.1% to 46.5% after installation in the garden-installed households, while the humidity was 38.9% to 44.9% in the non-installation households. The temperature range was 20.6 to 32.1 °C before installation, 26.0 to 29.7 °C after installation, and 28.7 to 30.4 °C in the non-installation households.

### 3.2. PM (Particulate Matter) Measurement Results According to Plant Installation

For the four households with indoor gardens, particulate matter was measured before and after the installation by dividing the living room and kitchen. The results are shown in Table 9. The concentration of the particulate matter in the living room and kitchen was 29.81 µm/m^3^ and 29.22 µm/m^3^, respectively, before the indoor garden was installed, while the concentration of particulate matter was reduced to 24.81 µm/m^3^ and 25.96 µg/m^3^, respectively, after the indoor garden was installed. After the indoor garden was installed, the particulate matter decreased by 16.79 µm/m^3^ in the living room and by 11.17 µm/m^3^ in the kitchen.

The result of a Wilcoxon rank sum test of the living room and kitchen showed a statistically significant difference when comparing before and after the installation of the indoor gardens.

In the living room and kitchen of all households, when comparing the time of the day, there was a statistically significant difference (Table 10).

### 3.3. IAQ Composite Index, Comfort Index, and Green Index Measurement Results

The pre-installation IAQ index ranged from 28.0 to 57.8 in the garden-installed households, with an average of 42.2. In addition, when plants were installed, the IAQ index ranged from 10.2 to 59.4 with an average of 37.9, and 12.9 to 38.9 with an average of 38.5 in the non-installation households. Figure 3a,b show that, when the indoor garden is installed, the average of the IAQ index is widened overall in the “good” section, as well as being widened in the “moderate” section. In addition, it was reduced in the “unhealthy sensitive group” section.

When comparing the values before the installation and after the installation of the plants, the increase in the “good” section of the IAQ index after installing the plants is thought to be due to the decrease in CO_2_ concentrations after plant installation.

The pre-installation comfort index ranged from 73.0 to 98.7 with an average 75.1 in the garden-installed households, and the comfort index ranged from 12.2 to 99.9 with an average of 72.7 when plants were installed. While the comfort index ranged from 26.0 to 99.9 with an average of 72.4 in non-installation households. In Figure 4a,b, comparing the comfort index before and after the installation of the indoor garden, the “uncomfortable” section was reduced and the “comfortable” section was increased after the installation of the indoor garden.

Furthermore, in the case of the garden-installed households, the “uncomfortable” section was reduced, and the “comfortable” and “very comfortable” sections were increased more than before. It is considered that the installation of plants influenced the temperature and humidity, affecting the comfort index of the occupants.

The green index before installation ranged from 50.5 to 74.0 in the garden-installed households with an average of 58.6, while the green index range was 18.1 to 74.5 with an average of 54.8 when the indoor garden was installed. In addition, the green index ranged from 21.2 to 75.1 with an average of 53.8 in the non-installation households. In Figure 5a,b, when sensors and plants were installed, the proportion of “unhealthy for sensitive groups” values decreased more than before installation, and the “moderate” and “good” sections increased.

On Figure 6, it showed IAQ index, comfort index and IAQ composite index.

The ratio of plants in the indoor space, however, should be considered according to individual preferences since there may be side effects depending on the indoor residents when the amount of plants is excessive.

## 4. Discussion

In the case of all of the households, the average humidity did not exceed 50%, which is the indoor humidity range recommended by American national standards institute (ANSI) / ASHRAE, and the temperature was higher than the 24 to 27 °C which is the summer temperature range recommended by ANSI/ASHRAE [23]. In some households, the humidity increased more after the installation of the indoor garden than before, which is believed to be due to the automatic humidity control of the installed Bio Wall [24]. Since the temperature and humidity and IAQ values change significantly from month to month, seasonal factors should be considered for a reasonable comparison between garden-installed and non-installation households.

The concentration of PM_2.5_ was decreased in the living room and kitchen after the indoor garden was installed. Lohr and Pearson first verified the fine particulate removal effects of plants in indoor spaces in 1996 [25], reporting that the presence of plants affects the reduction rate of particulate matter. Air purification of particulate matter by plants generally includes adsorption of indoor pollutants into the leaves of plants [26] and absorption of pollutants through plant pores [27,28]. In particular, particulate matter is known to be reduced through adhering to the wax layer on the leaf surface of a plant [25]. Although a correlation analysis on the amount of dust accumulated on the wax layers and leaf-surface cleaning and its frequency was not conducted in this study, some effects were still considered despite the low concentration of particulate matter in the households due to the participation of households with a high interest in plant cultivation. On the other hand, indoor plants could be one of the predictors of particulate matter through the effect of submicron particles and resuspension of soil particles. That is, more indoor plants in the residential space caused higher indoor particulate matter concentrations [29]. Therefore, further research should consider the possibilities that particulate matter could increase as a result of indoor plants.

In addition, CO_2_ showed a reduction in concentration in all households. This study considered influencing factors of indoor air quality such as time, season, and outdoor air quality in univariate ways. The correlation coefficients of PM_2.5_ concentration in the living room and kitchen were 0.60 and 0.70, respectively, indicating that indoor air quality was significantly affected by external influences. An increase in CO_2_ causes increased fatigue, decreased perception, and drowsiness, and a high concentration of CO_2_ tends to reduce work efficiency. It is also known that people experience headaches, dizziness, tiredness, and lack of concentration in environments with CO_2_ levels above 1000 ppm [30].

This study calculated the composite IAQ index based on the IAQ index and comfort index. In the case of particulate matter, there was no data for the upper limit for “risk” as regards effects on humans. It was calculated as being 70 µg/m^3^: a 15% increase of death risk rate according to the WHO recommendation standard, BI human body impact in the UK, and the Air Quality Index (AQI) of the US. Although the CO_2_ levels of the IAQ index should be calculated at a higher proportion because the CO_2_ reduction effect of indoor plants has already been verified in previous studies, the weights of PM_2.5_ and CO_2_ in the IAQ index were calculated as 1:1 in this study due to the importance of the health hazards associated with particulate matter in Korea.

Several studies have already shown that indoor plants improve the indoor air quality and reduce pollutants [31] and that they help control the humidity and temperature [32]; however, data on pollutants such as volatile organic compounds (VOCs) were not utilized in this study due to limitations of the current sensing technology on the market. Therefore, it is expected that IoT-based monitoring of various indoor air-quality issues will become available with the development of sensing technology in the future.

## 5. Conclusions

This study established and analyzed data based on particulate matter, carbon dioxide, temperature, and humidity sensors for new apartments with the aim of developing an IoT-based IAQ composite index using indoor gardens to protect the health of the occupants. After analyzing the data, the IAQ and comfort indices were calculated, and a comprehensive IAQ composite index was presented.

The result of investigating indoor pollutants when indoor gardens were installed showed that the installation of plants reduced some of the particulate matter and CO_2_ concentrations, which also affected the IAQ index. In the case of CO_2_, the garden-installed households showed lower concentrations than those without indoor gardens. In addition, in the case of particulate matter, there was no significant difference in the concentration between the garden-installed households and the non-installation households, while there was a difference in the concentration of particulate matter before and after installing the plants in the garden-installed households. According to a separate investigation of living rooms and kitchens, particulate matter was reduced in both spaces.

In the case of households with indoor garden installations, the IAQ index showed an increase in the “good” section when compared with before installation, and the “bad” section present before the installation disappeared. In the comfort index, the “uncomfortable” section that existed before installation then disappeared, and the average index level dropped considerably from “unhealthy for sensitive groups” to “good.” Overall, the IAQ composite index of garden-installed households decreased in the “good” section but did not influence the “bad” section. In addition, compared to the households without indoor gardens, the garden-installed households showed lower index ratings overall. The garden-installed households had no “bad” sections in the IAQ composite index after installing the plants, while the non-installation households had a “bad” section.

As a result, it was possible to observe the effect of improved indoor air quality with respect to some particulate matter and carbon dioxide and to utilize the composite index by developing indices representing IoT sensor-based indoor air quality and comfort. This study has limitations regarding representativeness since it included only limited items such as particulate matter, carbon dioxide, temperature, and humidity for the IAQ index and utilized data from only 16 spaces in eight households. Additionally, in the case of the composite index, the “uncomfortable” in the indoor garden installation group was higher than in the non-installed group due to seasonal differences (summer and winter) and short-term measurements of the background concentration. Furthermore, there were various factors which could have affected the results such as methods of ventilation in eight households after the indoor gardens were installed. In addition, the results of this study did not show a dramatic variation of the concentration of PM_2.5_ and CO_2_ according to season due to the low background concentration of PM_2.5_ and CO_2_.

On the basis of this research data, further research is necessary regarding the installation of indoor gardens in various indoor spaces by strengthening the framework of sensor-based IoT experiments and improving the performance of low-cost sensors.

## Figures and Tables

**Figure 1 ijerph-17-01867-f001:**
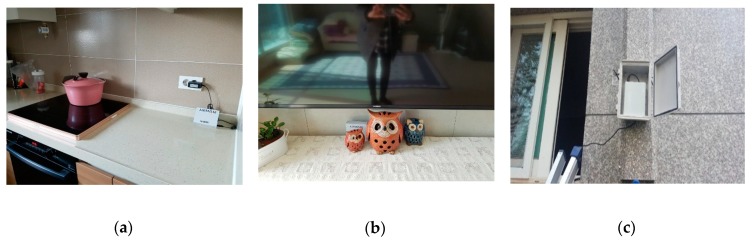
Details of sensor installation in the kitchen (**a**), living room (**b**), and outdoors (**c**).

**Figure 2 ijerph-17-01867-f002:**
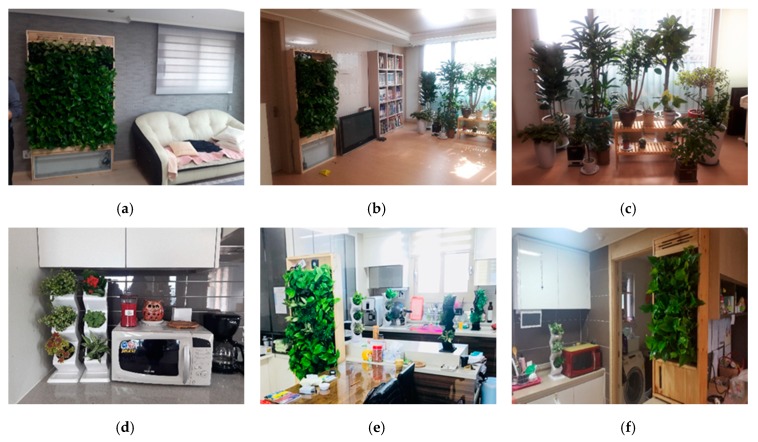
Details of indoor garden installation in the living room (**a**–**c**) and kitchen (**d**–**f**).

**Figure 3 ijerph-17-01867-f003:**
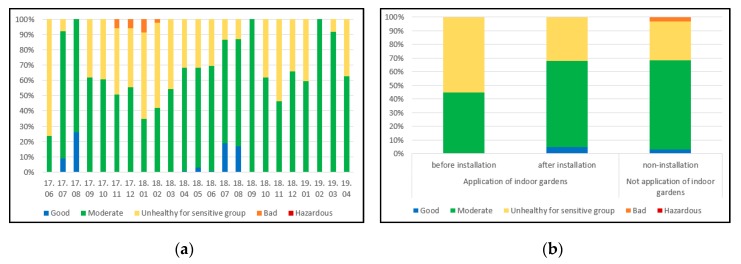
(**a**) IAQ index of measurement period; (**b**) IAQ index according to installation of indoor gardens. 17.—year 2017; 18.—year 2018; 19.—year 2019; Bottom line of numbers represents months in each year.

**Figure 4 ijerph-17-01867-f004:**
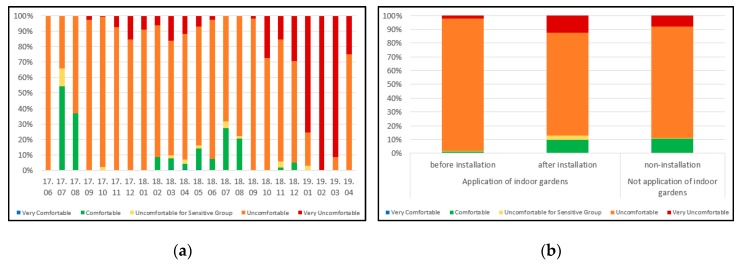
(**a**) Comfort index of measurement period; (**b**) comfort index according to installation of indoor gardens. 17.—year 2017; 18.—year 2018; 19.—year 2019; Bottom line of numbers represents months in each year.

**Figure 5 ijerph-17-01867-f005:**
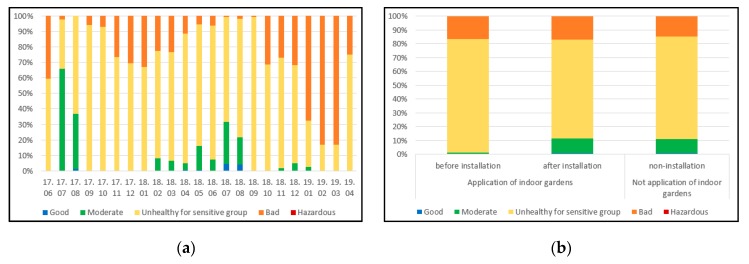
(**a**) IAQ composite index of measurement period; (**b**) IAQ composite index according to installation of indoor gardens. 17.—year 2017; 18.-year 2018; 19.—year 2019; Bottom line of numbers represents months in each year.

**Figure 6 ijerph-17-01867-f006:**
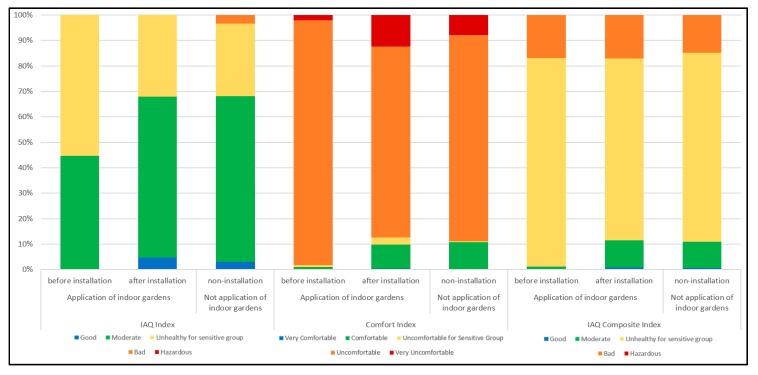
Index according to the installation of indoor gardens.

**Table 1 ijerph-17-01867-t001:** Details of characteristics of households.

Household	Move in	Number of People	Window Ventilation			Fan Ventilation		
			Number of Times	Period	Time (min)	Number of Times	Period	Time (min)
H1	2017 Dec.	2	2–3	Morning, afternoon, evening	15	2–3	-	10–20
H2	2017 Aug.	4	1	Morning, afternoon	10–20	2–3	Morning, afternoon, evening	120–180
H3	2017 Sep.	3	3–4	Morning, afternoon, evening	20	2–3	Morning, afternoon, evening	20
H4	2017 Mar.	4	2–3	Morning, evening	20–30	2–3	Morning, evening	30
H5	2017 Nov.	3	-	-	-	1	Morning, afternoon, evening	30
H6	2017 Oct.	4	2–3	Morning, afternoon, evening	30	3	Morning, afternoon, evening	-
H7	2017 Feb.	4	-	-	-	1	Evening	120–180
H8	2017 Feb	4	1–3	Morning, afternoon, evening	15–30	-	-	-

**Table 2 ijerph-17-01867-t002:** Specification of Internet-of-Things (IoT) device hardware.

Compose	Detail Content
PM_2.5_	0–1,000 µg/m^3^ (±15%)
CO_2_	0–2,000 ppm (±2%)
Temperature	−40–125 °C (±0.4%)
Humidity	0%–100%
Network	ETHERNET
Size (W × H × D)	110 × 90 × 33
Power	5V/1A
Operation Temperature	−10–50 °C

**Table 3 ijerph-17-01867-t003:** Scale of 1–100 points and color-coded index interval.

Index	A (0–20)	B (21–40)	C (41–60)	D (61–80)	E (81–100)
Sort	Good	Moderate	Unhealthy for sensitive groups	Bad	Hazardous

**Table 4 ijerph-17-01867-t004:** Integrated indoor air-quality (IAQ) index calculation.

Component	Good	Moderate	Unhealthy for Sensitive Groups	Bad	Hazardous
	A (0–20)	B (21–40)	C (41–60)	D (61–80)	E (81–100)
PM_2.5_ (µg/m^2^)	0–15	16–35	36–50	51–80	>81
CO_2_ (ppm)	0–450	451–700	701–1000	1001–3000	>3000

**Table 5 ijerph-17-01867-t005:** Classification of IAQ index.

Rating	Index	Score	Color
A	Good	0–20	Blue
B	Moderate	21–40	Green
C	Unhealthy for sensitive groups	41–60	Yellow
D	Bad	61–80	Orange
E	Hazardous	81–100	Red

**Table 6 ijerph-17-01867-t006:** Classification of indoor air comfort index.

Rating	Index	Score	Color
A	Very comfortable	0–20	Blue
B	Comfortable	20.1–40	Green
C	Uncomfortable for sensitive group	40.1–60	Yellow
D	Uncomfortable	60.1–80	Orange
E	Very uncomfortable	80.1–100	Red

**Table 7 ijerph-17-01867-t007:** Classification of IAQ composite index.

Rating	Index	Score	Color
A	Good	0–20	Blue
B	Moderate	21–40	Green
C	Unhealthy for sensitive group	41–60	Yellow
D	Bad	61–80	Orange
E	Hazardous	81–100	Red

**Table 8 ijerph-17-01867-t008:** Indoor air quality of households according to the installation of indoor gardens.

		Household	PM_2.5_ (µg/m^3^)	CO_2_ (ppm)	Tair (°C)	RH (%)
Installation of indoor gardens Jun 19 2017	Measurement before indoor plant	H1	17.4 ± 6.6 (7.0–33.0)	655.7 ± 111.1 (492.2–846.0)	27.4 ± 0.6 (26.2–28.5)	32.7 ± 1.9 (29.6–36.2)
		H2	16.8 ± 7.8 (3.5–38.4)	1185.4 ± 247.7 (614.0–1903.5)	27.1 ± 0.8 (25.7–29.6)	40.5 ± 2.9 (32.6–45.7)
		H3	22.0 ± 9.6 (3.9–50.2)	980.0 ± 280.8 (393.9–1713.5)	20.6 ± 1.5 (16.9–24.3)	39.1 ± 7.8 (14.3–45.1)
		H4	31.8± 10.1 (4.8–45.3)	653.9 ± 127.4 (471.0–847.4)	32.1 ± 2.4 (27.0–34.0)	40.3 ± 7.0 (28.6–49.4)
	Measurement after indoor plant	H1	15.8 ± 9.0 (1.0–45.4)	570.0 ± 129.3 (425.6–1015.4)	27.8 ± 2.4 (24.8–34.5)	37.1 ± 7.9 (18.8–56.9)
		H2	13.8 ± 6.6 (4.1–38.9)	1141.4 ± 373.2 (449.9–1946.3)	28.4 ± 2.2 (24.9–34.0)	42.8 ± 5.2 (21.7–51.4)
		H3	24.6 ± 13.0 (1.5–63.8)	832.4 ± 304.3 (442.1–1655.3)	26.0 ± 4.0 (15.7–34.1)	46.5 ± 5.9 (11.9–57.6)
		H4	14.2 ± 7.9 (4.6–53.7)	828.8 ± 209.5 (418.0–1531.1)	29.7 ± 2.7 (25.3–35.5)	37.6 ± 9.1 (22.7–56.4)
No installation of indoor gardens Jan. 11 2018		H5	15.1 ± 7.3 (1.9–42.5)	741.8 ± 187.9 (391.8–1223.6)	28.7 ± 2.4 (25.2–32.9)	44.9 ± 6.0 (29.7–54.6)
		H6	8.2 ± 5.7 (1.0–31.0)	1012.8 ± 350.3 (420.6–1930.0)	28.7 ± 2.4 (25.0–35.0)	40.4 ± 5.7 (21.5–51.9)
		H7	35.5 ± 64.5 (4.2–408.9)	983.6 ± 232.3 (448.0–1705.0)	30.2 ± 2.0 (25.6–34.4)	42.3 ± 7.4 (24.7–58.0)
		H8	18.8 ± 13.0 (1.3–63.3)	857.3 ± 323.1 (397.6–1623.2)	30.4 ± 3.1 (24.9–36.4)	38.9 ± 6.8 (22.3–55.6)

**Table 9 ijerph-17-01867-t009:** Concentration of particulate matter in living room and kitchen of all households.

Measurement Point	Measurement before Installation of Indoor Gardens (µg/m^3^)	Measurement after Installation of Indoor Gardens (µg/m^3^)	*p*-Value for Wilcoxon Rank Sum Test
Living room	18.82 (8.87)	16.94 (9.86)	0.0341
Kitchen	20.73 (9.69)	16.80 (10.44)	<0.0001

**Table 10 ijerph-17-01867-t010:** Concentration of particulate matter by the time of the day in all households.

Measurement Point	Day (µg/m^2^)	Morning (µg/m^2^)	Afternoon (µg/m^2^)	Evening (µg/m^2^)	*p*-Value
Living room	18.47 (31.46)	22.41 (39.60)	22.55 (42.05)	24.09 (43.33)	0.0217
Kitchen	13.86 (9.86)	17.67 (10.54)	15.91 (9.37)	17.63 (10.78)	0.0015

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
