# Peer review of "Evaluation of IAQ Management Using an IoT-Based Indoor Garden"

_ijerph, 2020, doi:10.3390/ijerph17061867_

Round 1

Reviewer 1 Report

The manuscript titled “Evaluation of IAQ management using an IoT-based indoor garden” documents an experiment conducted in indoor garden, and the objectives are (1) to demonstrate the effectiveness of garden in improving the indoor air quality, and (2) to propose the use of an IAQ composite index. Overall the study is interesting, but the authors will need to provide some further details on the experimental setups and index calculation. The authors will also need to reconsider the use of “smart” and “big data”.

Line 2: Please spell out “IAQ”. Line 18: Why the experimental field is “smart garden”? I did not see any significant differences between these experimental gardens and existing measurements. Lines 29-31: Not clear what “good”, “bad”, “uncomfortable”, and “comfortable” mean in the abstract. The abstract should be more self-explanatory. Line 59: Please provide more details/background of “sensor-based Internet-of-Things”. Lines 78–83: Do these eight apartments have similar ambient environment (vegetation, background concentration, etc.)? What is the setting of indoor garden (a picture/photo)? Is there any existing natural ventilation? According to Section 2.3, the setups of these plants are more like “indoor vegetation” instead of an “indoor garden”. Section 2.4: I don’t think the current analysis should be called “big data analysis” considering the amount of data collected (8 apartments). Lines 161-170: Besides the absorption, plants may have some negative impacts on PM concentration by changing the flow pattern, which may lead to even higher concentration level. See e.g., Quantifying the impact of urban trees on passive pollutant dispersion using a coupled large-eddy simulation–Lagrangian stochastic model. Building and Environment, 145, 33-49. https://doi.org/10.1016/j.buildenv.2018.09.014 Table 3: Are the differences statistically significant? How do they change with time (e.g., daytime vs. nighttime)? Lines 173-174: “based on the current maintenance standards and recommended standards”. References needed. Lines 187-189: The range (80-120) is different from that in Table 5 (36-50). For Table 5, what if the PM2.5 concentration is 36-50, while CO2 concentration is 0-450? Is it “good” or “unhealthy for sensitive groups”? Sections 3.3.3 and 3.4: Please provide the equations of the integrated IAQ index and indoor air comfort index. Equation 2: The left-hand-side is missing. Figures 1-3: Please explain why the “non-installation” case is better than the “before installation” case. Also, what is “not application”?

Reviewer 2 Report

The paper deal with the comparison of Indoor Air Quality in indoor environments with and without indoor gardens.
The topic is very interesting, the introduction is clear and well written. However, the paper presents some problems and should be improved.

2- Materials and methods
This section is too short. Much of the information needed to make the explained activity repeatable are missing.

The IoT should be better explained.The authors should better explain the convenience of the use of IoT technologies in comparison with traditional monitoring activities.

The case studies used should be more in the details described. A lot of information useful to better understand the research activity should be added.

The used sensors are described but it is not clear in which points of the kitchen and living room they are installed (it is different if a sensor is placed near a window or in the center of a room or in other positions).

The use of the indoor gardens should be better scribed adding more information (i.e. images and photos could be useful to better understand the impact and dimensions of the plants).

It is not clear if the monitored apartments are used during the monitoring activities and in this case it is necessary to specify the number of the persons, the occupancy profile, the profile of use of the building services (especially natural of mechanical ventilation systems), etc.
The indicators for the analysis of the IAQ are only mentioned but it is necessary to describe it in detail. The equations of the combined index and comfort index should be shown. It is also important to distinguish better the index selected from the scientific literature (with the relative references) and the index proposed by the authors.

3- Results
In this section, only the results of the monitoring activity should be reported. The information related to the classification of the values of IAQ indicators should be moved in "Materials and Methods", the analysis of results and eventually the proposal of a new index should be reported in in a new section "Discussion".

At which period are referred to the data "before indoor plant" and "after indoor plant" shown in Table 2?
In Table 3 the unit of the PM concentrations is lacking.

The results of the outdoor measures should be added and the differences between indoor and outdoor concentrations should be discussed. (For example in Europe, in the EN 16798-1:2019 the CO2 concentration is compared with outdoor concentration and the IAQ is calculated in relation to the increase of indoor CO2 concentration with respect to outdoor concentration).

Given the huge amount of data collected, and since plants are living organism and their behavior varying with the hours of the day and night, it would be very interesting to show for one or more days the difference between the trends of the pollutants concentrations in cases with and without the indoor gardens.

Reviewer 3 Report

1. The presentation of this manuscript can be further improved especially in the introduction and discussion sections.

2. More literature review is needed for both authors and readers' knowledge.

3. Further results on the IAQ management evaluation can be drawn.

4. The readability of all the figures is currently very poor. Image quality should be in an international standard.

5. Table 2 should fit on a single page.

Author Response

  1. The presentation of this manuscript can be further improved especially in the introduction and discussion sections.

Complemented.

  1. More literature review is needed for both authors and readers' knowledge.

Added.

  1. Further results on the IAQ management evaluation can be drawn.
  2. The readability of all the figures is currently very poor. Image quality should be in an international standard.

Revised.

  1. Table 2 should fit on a single page.

Revised. Also, Changed to Table 4.

Reviewer 4 Report

The paper titled: “Evaluation of IAQ Management Using an IoT-Based 2 Indoor Garden” shows real-time data and big data on PM2.5, CO2, temperature, and humidity collected through an IoT-based IAQ monitoring system new apartments divided into four indoor-garden-installed households and four households without indoor gardens.

The topic is worthy of investigation since the indoor air quality is having in the last decade an increasing relevance mainly in office building. In residential building the issues is less investigated and the paper can contribute in this field. However I suggest the following revision before publication:

In section 2 “Materials and Methods” please detail the sensor and indoor garden installation. This section is general, the experimental set-up is hard to understand. Eg. Position of sensors, dimension of the garden installation, typology of foliage etc. The results section must be integrated with graphs helping the readers to see and understand the difference between measurements. Regarding the comfort index there is no reference to the ISO 7730 and ISO 15251. Please argue. The conclusion section is general, please recap the main findings of the work supporting the representativeness of the results and replicability of the study.

Author Response

In section 2 “Materials and Methods” please detail the sensor and indoor garden installation. This section is general, the experimental set-up is hard to understand. Eg. Position of sensors, dimension of the garden installation, typology of foliage etc.

Added. Fig 1 and fig 2.

The results section must be integrated with graphs helping the readers to see and understand the difference between measurements.

Added

Regarding the comfort index there is no reference to the ISO 7730 and ISO 15251. Please argue.

Added Reference [20-22]

The conclusion section is general, please recap the main findings of the work supporting the representativeness of the results and replicability of the study.

Revised.

Round 2

Reviewer 1 Report

The manuscript has been substantially improved with clarification and elaboration supplemented in this revision. Here are two additional comments before the manuscript can be accepted for publication:

I am quite concerned about the use of “big data”. For my first comment, the authors stated that they have revised and deleted “big”, but “big data” still consistently appears throughout the revised manuscript. Please carefully check the manuscript to avoid the misuse of this term.

Please add the explanation of why the non-installation case is better than the before installation case in Section 3 or 4 to avoid confusion.

Author Response

I am quite concerned about the use of “big data”. For my first comment, the authors stated that they have revised and deleted “big”, but “big data” still consistently appears throughout the revised manuscript. Please carefully check the manuscript to avoid the misuse of this term.

Deleted “big”.

But, the “big” of line 62, 69, 70 and 71 is from references.

Please add the explanation of why the non-installation case is better than the before installation case in Section 3 or 4 to avoid confusion.

Complemented in “conclusion”.

Also, in the case of the composite index, the “uncomfortable” in the indoor garden installation group was higher than in the indoor ~ according to season due to low background concentration of PM2.5 and CO2 .

Reviewer 2 Report

The authors have adopted all the suggestions made, and the paper is now significantly improved.
In order to reach a version available for the publication, I suggest some minor corrections.

1- Page 5, line 179, I think that the number (2.4) of the section is lacking.

2- Section 2.4.2, please add the unit to the CO2 concentration ranges.

3- The term "Compose" in Tables should be replaced with more specific ones.

4- The households used as case studies (Tables 1 and 8) are called with the same "name"/letter of the rating classes (Tables 5,6,7). In order to avoid difficulties in the reading (especially of Tables), I suggest using different "names" for the case studies.

5- In Table 8, it could be useful to specify also the data of the measures before and after the installation of the plants. Furthermore, I believe that should be used the abbreviation Tair (air temperature) and RH (Relative Humidity) instead of TEMP and HUMI.

6- Please specify how the values of Table 10 have been obtained. Are they obtained from the mean of all the measures of all the case studies?

7- (facultative) Only for aesthetic purpose, I suggest that the letters (a) and (b) in the Figures 4 and 5 could be moved above the graphs in order to better separate them from the captions.

Author Response

  • Page 5, line 179, I think that the number (2.4) of the section is lacking.

Revised. 2.4 Calculation of an indoor environment comprehensive IAQ index

  • Section 2.4.2, please add the unit to the CO2 concentration ranges.

Revised. Add the unit (ppm).

  • The term "Compose" in Tables should be replaced with more specific ones.

Revised.

  • The households used as case studies (Tables 1 and 8) are called with the same "name"/letter of the rating classes (Tables 5,6,7). In order to avoid difficulties in the reading (especially of Tables), I suggest using different "names" for the case studies.

Revised. The households of case studies were replaced H1 to H8.

  • In Table 8, it could be useful to specify also the data of the measures before and after the installation of the plants. Furthermore, I believe that should be used the abbreviation Tair (air temperature) and RH (Relative Humidity) instead of TEMP and HUMI.

Revised. Tair and RH.

  • Please specify how the values of Table 10 have been obtained. Are they obtained from the mean of all the measures of all the case studies?

Revised. In the living room and kitchen of all households,~

Add “all households” to the titles of table 9 and 10.

7- (facultative) Only for aesthetic purpose, I suggest that the letters (a) and (b) in the Figures 4 and 5 could be moved above the graphs in order to better separate them from the captions.

Revised.